# Deimination, Intermediate Filaments and Associated Proteins

**DOI:** 10.3390/ijms21228746

**Published:** 2020-11-19

**Authors:** Julie Briot, Michel Simon, Marie-Claire Méchin

**Affiliations:** UDEAR, Institut National de la Santé Et de la Recherche Médicale, Université Toulouse III Paul Sabatier, Université Fédérale de Toulouse Midi-Pyrénées, U1056, 31059 Toulouse, France; julie.briot@inserm.fr (J.B.); michel.simon@inserm.fr (M.S.)

**Keywords:** citrullination, post-translational modification, cytoskeleton, keratin, filaggrin, peptidylarginine deiminase

## Abstract

Deimination (or citrullination) is a post-translational modification catalyzed by a calcium-dependent enzyme family of five peptidylarginine deiminases (PADs). Deimination is involved in physiological processes (cell differentiation, embryogenesis, innate and adaptive immunity, etc.) and in autoimmune diseases (rheumatoid arthritis, multiple sclerosis and lupus), cancers and neurodegenerative diseases. Intermediate filaments (IF) and associated proteins (IFAP) are major substrates of PADs. Here, we focus on the effects of deimination on the polymerization and solubility properties of IF proteins and on the proteolysis and cross-linking of IFAP, to finally expose some features of interest and some limitations of citrullinomes.

## 1. Introduction

Intermediate filaments (IF) constitute a unique macromolecular structure with a diameter (10 nm) intermediate between those of actin microfilaments (6 nm) and microtubules (25 nm). In humans, IF are found in all cell types and organize themselves into a complex network. They play an important role in the morphology of a cell (including the nucleus), are essential to its plasticity, its mobility, its adhesion and thus to its function. There are six types (Types I–VI) of IF proteins, each belonging to a superfamily and classified according to the origin of the cell in which they are expressed (Table 1) [1,2].

In all cases, IF organize themselves in space according to a well-described process that allows their polymerization and assembly to form a dynamic three-dimensional network [3,4]. For example, keratins (Types I and II), encoded by more than 50 distinct and functional genes, are expressed in epithelial cells (epidermis and pulmonary, uro-genital, intestinal or oral epithelia) and in skin appendages (nails, hair and teeth) [5]. One acidic keratin (Type I) and one basic keratin (Type II) combine to form a heterodimer, then an anti-parallel tetramer and finally polymerize into unpolarized IF [3,6]. IF are found throughout the cytosol, between the nucleus and the plasma membrane, with the exception of lamins, which constitute the *lamina nuclea*, i.e., the nucleus IF network located under the nuclear envelope. IF are able to assemble spontaneously in vitro. The very stable assembly of constituent subunits has been observed by electron microscopy after negative staining [7].

IF proteins interact with and are regulated by associated proteins called IFAP (intermediate filament associated proteins), such as filaggrin, trichohyalin, plakin, plectin and syncoilin [8,9,10,11,12,13]. IFAP allow IF to interact with desmosomes, hemidesmosomes and the nuclear membrane. In the epidermis, filaggrin is thought to promote the aggregation of keratin IF and could participate in their condensation [14,15]

The IF proteins and, in particular, their non-helical head and tail domains located at their *N*- or *C*-terminal ends undergo numerous post-translational modifications [16], such as phosphorylation, glycosylation or sumoylation, which regulate their dynamic remodeling (Table 2).

One of the post-translational modifications targeting the non-helical ends of IF proteins is deimination (or citrullination) (Figure 1a). Deimination is an irreversible reaction, catalyzed by a family of calcium dependent enzymes, called peptidylarginine deiminases or PADs (EC3.5.3.15) [8,30,31,32] (Figure 1b). This enzymatic reaction corresponds to the transformation of the guanidino group of an arginine residue (positively charged) into a ureido group, resulting in a citrulline residue (neutral), within a substrate protein (or a peptide). Some IFAP have also been shown to be modified by deimination (Table 2 and Figure 1c,d). This modification regulates their sensitivity to proteolysis [8,30,31,32,33] and their ability to be cross-linked by transglutaminases [34,35,36], which in turn modulates their structural organization following the loss of positive charges [13]. The first proteins demonstrated to be deiminated (or citrullinated) by PADs, identified by two-dimensional gel electrophoresis and immunodetection using a specific anti-modified citrulline (AMC) antiserum [17], were abundant proteins. They mainly corresponding to either constituents of IF or IFAP (Table 1). 

Here, we propose an update on the deimination of IF proteins and IFAP, with a particular focus on keratins, filaggrin, trichohyalin and keratinocyte differentiation. We also describe the importance of deimination and PADs in several physiological (differentiation, embryogenesis, gene regulation and inflammation) and pathological (rheumatoid arthritis, multiple sclerosis, Alzheimer’s disease and cancers) contexts, where they have been widely studied for several decades [8,9,12,30,31,32].

## 2. Five Human PADs, Their Differential Expression, Localization and Implication

In humans, there are five *PADI* genes located at a single locus (1p35–36) [40] and encoding five distinct PADs (PAD1–4 and PAD6). The amino acid sequences of the five PADs (and the nucleotide sequences of the *PADI* genes) are conserved and present high levels of homology, ranging from 59% to 71% [8,30,31]. They have differential expressions depending on the tissue or cell types [40,41,42], distinct substrates and specific calcium sensitivities [39,43,44]. The fine regulation of the expression of each *PADI* gene has been extensively described in keratinocytes, where it has been shown to be under the control of calcium, cell confluency or vitamin D treatments [45,46,47], as well as in cancer cell lines stimulated by all-trans-retinoic acid [48]. PADs appear as “multilocked” enzymes [47]. The enzymes (and their genes) are controlled at multiple levels (transcriptional, translational and post-translational, including auto-deimination) [47,49,50,51]. Non-physiological calcium concentrations or the use of ionophores are required to induce a high deimination rate, named “hypercitrullination” ([48] and part 5).

PAD1 is mainly expressed in the uterus and epidermis [41,52]. It has been detected, in particular by our team, in the cytoplasm of all human keratinocytes. In more differentiated living cells, the granular keratinocytes, PAD1 is associated with IF of keratins and located at the level of granular protein structures called keratohyalin granules [30,39,40,41] (Figure 2a–c). The keratohyalin granules have recently been described as dynamic structures, corresponding to liquid-like condensates able to undergo liquid-liquid phase separation during keratinocyte differentiation [53]. Their content, including filaggrin, is released in the transitional cell, just under the *Stratum corneum*, the most external part of the epidermis. Filaggrin is an IFAP that interacts with keratins and is a major protein for keratinocyte differentiation [9,10,11] (see also Section 4). Filaggrin is widely studied in the context of atopic dermatitis, because loss-of-function mutations of its encoding *FLG* gene are a major genetic risk factor for this common skin disease (also known as eczema) [12,54] (OMIM #605803 and #135940). Filaggrin is a substrate of PAD1–3, PAD1 showing the best in vitro capacity to modify this IFAP (Figure 1d) [39,41]. In human epidermis, PAD1 is also localized in all layers of corneocytes, anucleated and flattened cells derived from the differentiated keratinocytes and constituting the *Stratum corneum*, the protective horny layer of the epidermis. PAD1 is immunodetected at the level of the intra-corneocyte matrix, which is mainly made up of aggregated keratins [30,39,41] (Figure 2c). Recently, we showed, in a three-dimensional reconstructed human epidermis, that keratinocytes are able to adapt, in the absence of any other types of cells, to atmospheric dryness (relative humidity of 30–50%). Combined with an increased number of corneocyte layers, large amounts of PAD1 protein and mRNA and a markedly increased deimination in the *Stratum corneum* were observed [33]. Furthermore, in the context of tumorigenesis, PAD1 is significantly modulated and could be used as a biomarker [55,56]. For example, in triple-negative breast cancer, overexpression of PAD1 seems to be involved in the epithelial–mesenchymal transition and could promote metastasis [56].

PAD2 isotype is ubiquitous. It was originally described and purified from the skeletal muscle of rabbits [57,58]. It is the only PAD expressed in the central nervous system [42], where it is widely studied in the context of neurodegenerative diseases such as multiple sclerosis and Alzheimer’s disease [59,60] and in the glaucomatous optic nerve [61]. PAD2-deficient models (knockout mice or deficient human models obtained using RNA interference) demonstrate its involvement in inducing deiminated proteins in the nervous system, especially in astrocytes [60,61,62]. PAD2′s role in keratinocytes [42] remains to be deciphered since its absence in knockout mice did not induce any clear effects at the epidermal level [63]. In addition, PAD2 does not appear to be essential for the in vitro establishment of a human reconstructed epidermis, its mRNA being undetectable in this organotypic model [33]. In the context of Alzheimer’s disease, the glial fibrous acidic protein (GFAP) and vimentin are identified as PAD2 major substrates, at the level of the hippocampus, where PAD2 is also immunolocalized [60]. Most recently, PAD2 was described in oligodendrocytes, where it plays a major role during cell differentiation and in the establishment of axonal myelination [64]. 

PAD3 is specifically expressed in granular keratinocytes of normal human epidermis, where it co-localizes with filaggrin on keratohyalin granules (Figure 2d,e) [30,39,42], and in the hair follicle [30,39,65], where it plays a major role in the regulation of trichohyalin, a filaggrin-like protein. Trichohyalin was the first IFAP described as strongly deiminated in vivo [24]. In 2016, homozygous or heterozygous composite missense mutations of the *PADI*3 gene were described as responsible for uncombable hair syndrome (UHS), a rare hair shaft dysplasia affecting young children, also known as *pili trianguli et canaliculi* [30,66] (Table 3). The syndrome is associated with a deformation of the hair shaft, without any notable fragility. This abnormal morphology of the hair shaft, observable by scanning electron microscopy, is used to establish the diagnosis. This abnormal phenotype has also been observed on dorsal hair and vibrissae of mice deficient for *Padi*3 [30,66]. In this later study, a metabolic pathway was proposed in the morphogenesis of the hair follicle. Three major proteins are involved: the trichohyalin IFAP and two post-translational modification enzymes acting on it, PAD3 and transglutaminase-3 (transglutaminases catalyze cross-links between glutamine (Q) and lysine (K) in polypeptidic substrates [66]). In 2019, distinct heterozygous mutations also affecting the *PADI*3 gene were described as associated with another disease affecting the hair, namely the central centrifugal cicatricial alopecia (CCCA) [30,67] (Table 3). This alopecia (hair loss) is often associated with peri-follicular inflammation; it is common in African women or in women of African origin with a prevalence of ~5.6% [30,67]. This disease seems to be partly linked to excessive hair styling in the affected populations. The molecular mechanisms altered by these mutations have not been clearly deciphered. To date, the activity of the corresponding recombinant mutated forms of PAD3 is disrupted or even absent [30,66,67]. A comparative analysis by RNA sequencing showed the downregulation of the expression of a large panel of genes encoding keratins (more than 20 with a fold change of -8 or less) and proteins associated with keratins (nearly 70 with a fold change of -8 or less). Among them, five genes (*KRT*81, *KRT*83, *KRT*86, *KAP*10.2 and *KAP*12.1) showed a significant downregulation confirmed by quantitative RT-PCR in the skin of patients as compared to healthy individuals [67]. At the same time, the expression of filaggrin was reduced by a factor of 3 while those of trichohyalin and trichohyalin-like were markedly reduced by factors of 15.8 and 13.3, respectively [67]. Moreover, another study has shown, using human embryonic neural cells, the involvement of PAD3 in association with the apoptosis inducing factor (AIF) in an apoptotic pathway, independently of caspase-3 activation [68]. 

PAD4, initially described in the rat (also named Pad/PAD type 5) [77], is mainly expressed in hematopoietic cells, where it is translocated to the nucleus under retinoic acid treatment [48,78]. It is the only isotype to carry a functional canonical nuclear localization signal [78]. In the nucleus, PAD4 is able to act on histones and plays an essential role in epigenetics and transcription regulation via the deimination of various targets including p53, histones and lamins. PAD4 is able to catalyze the deimination of arginines, and also of methylated arginines (in proteins or peptides) as demonstrated for histone methyl-arginines [71,72]. PAD4 is widely studied in the context of autoimmune diseases, such as rheumatoid arthritis [31,73,79,80,81] or lupus [82,83], diseases for which genetic links have been found, some haplotypes carrying an increased risk of developing the disease (Table 3). In the case of lupus, the authors studied nine single nucleotide polymorphisms (initially described as haplotypes associated with rheumatoid arthritis [73]), located all along the *PADI*4 gene, in the coding or non-coding regions. They identified a single polymorphism (rs1635564) of *PADI*4 associated with an increased risk of lupus erythematosus and nephritis [82]. In 2003, Kazukiho Yamamoto’s team provided the first demonstration of a genetic link between *PADI*4 haplotypes and the severe autoimmune disease, rheumatoid arthritis [73]. This major study has certainly boosted research studies on PADs (especially PAD4) and deimination. PAD4 has been implicated in several inflammatory and wound healing contexts [31,63,79]. PAD4 is also believed to play a major role during neutrophil extracellular trap activation and release (NETosis) [84,85,86], an innate defense process associated with neutrophil death during infection by bacteria and other pathogens. During NETosis, the neutrophils extracellularly release their relaxed nuclear chromatin together with cytoplasmic proteins, thus forming a trap (like a net) against the pathogens. PAD4 is necessary for chromatin relaxing via a hypercitrullination mechanism affecting histone H3, among others [84,85,86]. However, the involvement of PAD4 during NETosis is controversial, since it would be favorable to, but not absolutely necessary for, NETosis, at least in a model where NETosis was induced by *Candida albicans*, while deimination of histone H3 is effective [87]. Furthermore, induction of deimination by treatment of neutrophils with a calcium-ionophore induces dissociation of the IF of the *lamina nuclea* by a calpain-mediated proteolysis, which leads to complete dislocation of the nucleus [88]. Previously, the deimination of lamin C had already been observed in a cell model of induced apoptosis, where nuclear fragmentation was also described as dependent on PAD4 [23]. Therefore, in the nucleus of neutrophils, PAD4 catalyzes the deimination of the numerous nuclear proteins involved not only in its structural organization (lamins) but also in the chromatin organization (histones) or in gene regulation via p53, for example [86]. In the joints of rheumatoid arthritis patients, PAD4 and PAD2 mainly act on extracellular targets, the first described being fibrin (α and β chains). The deiminated fibrin is recognized by autoantibodies highly specific to rheumatoid arthritis and named anti-citrullinated protein autoantibodies (ACPA) [79,89,90]. Recently, PAD4 has been detected at the cell membrane of human neutrophils, and PAD2 associated with extracellular sub-fractions [91,92]. In fact, Pad2 seems to be associated with small extracellular vesicles, called exosomes, isolated, for example, from the plasma of *Alligator mississippiensis* [92]. Among about thirty deiminated proteins, deiminated keratins K14, K19 and K20 have also been shown to be associated with plasmatic exosomes, together with one IFAP, deiminated desmoplakin [92]. 

PAD6 is the last PAD to have been described. Originally, it was identified by a proteomic approach from oocytes of mice [74]. It was then named “ePad” for “embryonic Pad” acting in the remodeling of the cytoskeleton forming the lattice, an embryonic cell structure made up of microtubules [74,93]. In humans, the *PADI*6 gene was described in the context of a more global analysis of the 1p35–36 locus carrying the five *PADI* genes [40]. Its full cDNA was sub-cloned from an ovarian bank and it seems to be mainly expressed in ovary, testis and peripheral blood leucocytes [40]. Similar to the other *PADI* genes, *PADI*6 carries 16 exons, encoding a large protein of 694 amino acids, compared to 663–665 amino acids for the other human PADs [40]. However, its catalytic activity remains to be demonstrated [40,74,79,93]. PAD6 is clearly the most divergent isotype of the five PADs [31,40]. In 2016, two teams identified mutations in the *PADI*6 gene associated with early female infertility observed during medically assisted reproduction techniques [75,76] (Table 3). They are homozygous or composite heterozygous recessive mutations. These mutations (missense or inducing a stop codon) lead to a complete loss of PAD6 [75,76] and an early arrest of embryogenesis from the third day post-fertilization, as observed by time-lapse microscopy [75]. *PADI*6 has therefore become an important gene in the screening of early infertility post-fertilization [76].

## 3. Deimination of IF Proteins, Impact on Their Polymerization and Solubility

The first deiminated (or citrullinated) proteins identified under physiological conditions were trichohyalin from the hair follicle and filaggrin, keratin K1 and keratin K10 from the normal epidermis [17,18,24]. Thereafter, abundant proteins, such as myelin basic protein (MBP), fibrin, vimentin and GFAP, were also demonstrated to be targets of this post-translational modification, in several pathological contexts, such as multiple sclerosis, rheumatoid arthritis and Alzheimer’s disease [27,89,94,95,96].

In 1989, Chikako Sato’s team demonstrated for the first time the huge impact of deimination on the polymerization of vimentin IF induced by a purified active PAD [20]. This suggested a major role of deimination in the remodeling of IF in vivo. The solubility of the deiminated vimentin was observed to be increased compared to that of the unmodified protein, which remained stable and formed IF that were clearly visible under transmission electron microscopy after negative staining. The deiminated vimentin was no longer able to form stable IF. This work also showed that deiminated vimentin was no longer phosphorylated by protein kinases. In a mouse model of eye injury caused by an alkaline treatment at the eye surface, many deiminated proteins were detected following the trauma. Among them, GFAP and vimentin were more easily extractable in a low ionic strength buffer when deiminated. Once again proteins appear to be more soluble when they are deiminated [97]. This suggests that polymerization of IF is less efficient when their constitutive subunits are deiminated. Conversely, inhibition of deimination, by an irreversible chemical inhibitor of PADs, Cl-amidine (see also Section 5), stabilizes the vimentin IF in embryonic neural cell apoptosis induced by a calcium burst under thapsigargin treatment [68]. The increase in calcium concentration drives the depolymerization of vimentin and improves its extraction. This effect is markedly reduced by the PAD inhibitor. 

Keratins (at least K1) exhibit deimination sites in their variable regions V1 and V2, as initially demonstrated in the epidermis of mice (Figure 1a) [19,98,99]. In normal human epidermis, deiminated keratin K1 appears in the most superficial corneocytes, where only PAD1 is detected (Figure 2c) [19,39,98,99]. Tissue distribution of deiminated keratin K1 is largely disturbed in the epidermis of patients with bullous congenital ichthyosiform erythroderma (BCIE), a skin disease where the keratin IFs appear to be aggregated [98]. From purified human cornified envelope (insoluble cross-linked lipid-protein supramolecular structure surrounding corneocytes), we identified six deiminated keratins (three Type I keratins, i.e., K14, K16 and K17, and three Type II ones, i.e., K1, K2 and K6B) and characterized, at least in part, their deimination sites [30]. The deimination sites (1–2 per keratin) are mainly localized in their head and tail domains, in accordance with the fact that PADs preferentially target sites in the non-helical regions of proteins, and in particular in the V1 and V2 regions essential for the dimerization, tetramerization and therefore polymerization of IF [1,30] (Figure 1a). 

All these examples, in many cell types (keratinocytes, nerve cells, etc.) tend to demonstrate the essential role of deimination and PADs in the organization of IF, promoting their solubility and thus their dissociation.

## 4. Deimination of IFAP, Impact on Their Proteolysis

Deiminated filaggrin was first demonstrated in the epidermis of rats, and later in humans, by two-dimensional gel electrophoresis analysis [17,18,19]. The migration of the deiminated proteins during one- or two-dimensional gel electrophoresis is highly disturbed compared to the unmodified control (even in denaturing (SDS) and reducing (beta-mercaptoethanol) conditions). This was observed for filaggrin, trichohyalin and other proteins [19,39]. The apparent molecular mass change can reach several tens of kDa (as illustrated for filaggrin on Figure 1d) and the isoelectric point can vary by several pH units [18,19,96]. For instance, the various deiminated forms of human filaggrin exhibit isoelectric points varying from ~6 to 8, their two-dimensional electrophoretic profile forming a typical “comma” shape [18,19,96]. One singularity of filaggrin, and several other substrates of PADs, is its high content of arginine residue, and therefore potential deimination sites (filaggrin contains 11.2% arginine, 9.2% GFAP and 9.2% vimentin). This may partly explain the “comma” shape, linked to a large loss of positive charges. We demonstrated that the solubility of filaggrin is also greatly increased when it is deiminated [33]. Massive deimination of filaggrin is favorable to its complete proteolytic breakdown, leading to the production of free amino acid derivatives, a key process for the maintenance of tissue hydration [33]. We also identified a chemical activator of PADs, aceffyline, a xanthine derived from caffeine that promotes the deimination of filaggrin and consequently cornified epithelium hydration [100,101], as also observed by others [102]. Filaggrin is a basic protein that interacts with keratin IF in the upper epidermis to promote their aggregation [103]. The deiminated filaggrin (less positively charged) detaches from keratin IF and becomes more accessible to cytosolic proteases such as calpain-1 and bleomycin-hydrolase, two enzymes that cleave deiminated substrates with a higher efficiency [33,34,104]. In 1993, Peter Steinert’s group hypothesized, in the so-called ionic zipper model, that filaggrin combines and stabilizes keratin IF, simply because it displays many positive charges dispersed all along the molecule whereas keratins are negatively charged [105]. In this case, the zipper is in “closed mode”. We propose that the loss of all (or some) positive charges induced by its deimination leads filaggrin to become neutral (or even acidic), to gradually dissociate from keratins, to solubilize and to become more sensitive to peptidases. In this case, the zipper is in “open mode”.

The modulation of proteolytic sensitivity by deimination has been notified for other deiminated proteins, including MBP, which is better proteolyzed by cathepsin D after deimination [106]. The deiminated form of fibrin is also less efficiently cleaved by thrombin. This may participate in pro-inflammatory processes in rheumatoid arthritis [107,108]. An atypical organization of the fibrin network in patients with rheumatoid arthritis has recently been characterized by a scanning electron microscopy approach, using fibrin isolated from blood plasma [109]. In this study, the amount of deiminated fibrin was significantly higher in patients than in healthy controls, but this observation was not disease-specific, as already published [79]. Deimination could also regulate the proteolysis of some cytokines [110]. In the case of interleukin-8, the deimination of an arginine at the *N*-terminal end (Arg-5) decreases its proteolysis by thrombin and plasmin, reduces its binding to glycosaminoglycans (heparin and heparan sulfate) and shows a lower chemotactic activity against poly-nuclear neutrophils [110]. The effect of deimination on proteolysis could also occur for some serpins, key inhibitors of many serine or cysteine peptidases. Serpins carry a variable reactive center loop (RCL) able to specifically interact with the active site of the cognate peptidases [111]. Paul Thompson’s team demonstrated the inhibition of the activity of several deiminated serpins (such as antithrombin and antiplasmin). In particular, the deiminated serpin F2 (antiplasmin a2; Arg376Cit) no longer acted on plasmin, unlike the mutated form Arg376Lys, which remained active. Unlike positively charged arginine (or lysine), citrulline would not be able to interact in the active site of the peptidase [112]. A multiple alignment of 13 human serpin sequences demonstrates this could potentially concern many serpins (at least B2, B5, B6, B8, B12, C1, G1 and F2). Those members of the subgroup of serpins carry an arginine at the P1 position in their reactive center loop (Figure 3). Serpins play a major role in many processes related to inflammation, coagulation, complement and epidermal differentiation (desquamation and extracellular matrix). Such a mechanism of regulating proteolysis via the deimination of the peptidase inhibitors or peptidase substrates involved therefore seems particularly interesting and should be better deciphered in the future in physiological and pathological contexts.

In addition to its effect on the sensitivity to proteases, deimination also plays a role in sensitivity to transglutaminases, another family of post-translational modifying enzymes, carrying calcium dependent crosslinking catalytic activities [36,113]. During epidermal differentiation, transglutaminases cross-link proteins, including keratins and IFAP, to the cornified envelopes of corneocytes, and play an important role in the morphogenesis of the hair follicle [24,25]. Trichohyalin, a major IFAP of the hair follicle, is also highly deiminated. It holds the record of 23.9% of arginines in its primary sequence (377 arginines). Its high deimination rate leads to a decrease in its structural organization with, in particular, an unfolding of its α-helix domain (at least the recombinant form of domain-8), which increases its solubility and induces a greater sensitivity to the transglutaminase-3 activity [13,25]. An increased sensitivity to transglutaminase 1 or 3 has been described by our group for other deiminated proteins related to filaggrin and trichohyalin, i.e., hornerin and filaggrin-2 [9,114,115]. 

Thus, it appears that deimination modulates the organization of IF by acting directly on the head and tail domains of their constituents (such as vimentin, keratins and lamins). Deimination also acts on the repeated domain of some IFAP (as illustrated here for filaggrin and trichohyalin). Deimination induces a loss of positive charges, which causes drastic structural changes and alterations of the IF and IFAP organization, and thus modulates the solubility and accessibility of these proteins to, in particular, peptidases and transglutaminases.

## 5. Hypercitrullination or Hypocitrullination 

Hypercitrullination (or hyperdeimination) of a large panel of proteins was recently described in works concerning NETosis and analysis of deimination in cellulo induced by various exogenous stimuli. In many cases, PADs are not necessarily active/functional but rather appear as “multilocked” and require stimulating agents, such as differentiation, cell confluence, calcium, or retinoic acid to name but a few [47,48]. Calcium-selective ionophores, such as ionomycin, have been extensively used in various cellular models to induce hypercitrullination, although they are not always sufficient [86,116]. In monolayers of cultured human keratinocytes, we never observed endogenous deiminated proteins after 2 h of treatment with 5 μM of ionomycin, whereas PADs were immunodetected and PAD activity could be evidenced in vitro using an appropriate buffer (unpublished data). Other stimuli, such as phorbol esters or lipopolysaccharides, can induce “hypercitrullination” of some histones in neutrophils or monocytes, but without modification of a wide range of proteins [85,116,117]. On the other hand, the use of perforin or leukotoxin A, proteins forming pores in the cell membrane, is effective in inducing a massive hypercitrullination (like a storm) affecting a very large number of targets (beyond histones), including constituents of IF such as vimentin, lamin B1 and other cytoskeleton components, including tubulin beta-chain and actin [116,117]. 

Conversely, hypocitrullination and its effects could be analyzed using specific irreversible chemical inhibitors of PADs, especially the pan PADs inhibitor Cl-amidine and its derivatives [118,119,120], and knockdown mouse models [86]. For example, the absence of *Padi*4 causes hypocitrullination of histones and a lower efficiency in inducing NETosis [86]. Cl-amidine-induced hypocitrullination effects were initially observed in a human osteosarcoma cell line (U2OS) where some key autophagy proteins, such as LC3/Sestrin2/AHF4, were induced at the transcriptional level [118,119]. In a model of human epidermis reconstructed in 3D, we demonstrated that Cl-amidine, when it induces a decrease of around 50% in the deiminated protein rate, leads to a modulation of the autophagic pathway involved in keratinocyte differentiation [119,121]. An electronic microscopy approach revealed a decrease in the number of corneocyte layers, a higher frequency of transitional cells, a huge accumulation of heterogeneous vesicles in the cytoplasm of granular keratinocytes, sometimes around the nucleus and, occasionally, some autophagolysosomes and clusters of mitochondria. In this model, a significant decrease in profilaggrin (RNA and protein levels) was measured and a lower number of cell layers exhibiting transglutaminase activity was detected in situ. All these modifications, induced under hypocitrullination, demonstrate the importance of deimination and PADs (PAD1 and PAD3) during the keratinocyte terminal differentiation program at the transition between the granular layer and the *Stratum corneum* [119].

Finally, in vivo hypocitrullination can be exemplified for human pathologies by the effects of mutations of *PADI*3 and *PADI*6 genes on IF, IFAP and, more generally, on cytoskeleton proteins [66,67,75,76,122]. For *PADI*3 mutations, a significant decrease in the expression of many markers of keratinocyte differentiation in the hair follicle, in particular keratins and their associated proteins, was observed, as mentioned above [66,67] (see also Section 2). For *PADI*6 mutations inducing early infertility, this affects the expression of certain genes essential for cytokinesis during the early stages of embryogenesis, such as that of cofilin-1 (*CFL1*), which is involved in the organization of the actin cytoskeleton [75,76,122].

## 6. Proteomic Approaches of “Citrullinomes”

Researchers have long struggled to analyze deimination by classic proteomic approaches, since it induces a change in net charge but only a small gain of mass (~1 Da). In addition, the modification of an arginine to citrulline is not taken into account in the Mascot-type proteomic analysis and it can be easily confused with deamination (identical mass change of ~1 Da) of asparagine (N) and glutamine (Q). In recent years, super-resolution and specific proteomic approaches have been developed to identify the deiminated substrates and precisely characterize the sites of deimination. They are mostly based on the use of various additional and specific chemical groups derived by reacting 2,3-butanedione with citrullines [123,124]. These additional groups produce a significant change in mass, from 50 to 200 Da, which is more detectable by mass spectrometry. Chemical groups derived from phenyl-glyoxal (named probe-PG) were also developed, in particular to precisely identify the sites of deimination on substrates in complex biological samples [125,126]. 

Thus, many “citrullinomes” (lists of deiminated proteins) are now available, some from complex biological samples of healthy individuals or patients [30,112,115,126,127,128,129,130,131]. However, it is important to remember that the deimination can be induced by heating proteins in the presence of urea (homocitrullination) or modulated under the effect of bicarbonate [132]. For a long time, the components of IF (keratins and vimentin) and certain IFAP (trichohyalin and filaggrin) have been widely described as targets of PADs in vivo (Table 1). However, they certainly correspond to “the tip of the iceberg”, and many additional deiminated proteins will no doubt be identified and should be integrated in biological networks. As an example, a proteomic approach made it possible to describe, for the first time in humans, the presence of deiminated proteins in healthy and pathological heart tissue [130]. Vimentin (at least one peptide), was once again shown to be deiminated. Above all, this was the first work that identified deimination sites, beyond IF, on cytoskeleton proteins (actin, myosin, tropomyosin, troponin and filamin C). Thus, deimination could influence the calcium-dependent contractibility of cardiomyocytes or other cardiovascular cells acting on the actin/myosin interaction process. The deimination of myosin modulates its ATPase activity and therefore its efficiency during sarcomere contraction. Recently, in the context of neurons and multiple sclerosis, beyond neuro-filaments (NEF-L and NEF-H), MBP and GFAP, more than 20 new cytoskeletal proteins (tubulin, gelsolin, etc.) were also identified as deiminated [131]. The authors found other new deiminated proteins, some essential to the functional unit of the synapse, such as neurogranin, and some more ubiquitously expressed, such as the junctional proteins ZO-1, ZO-2 and alpha-2-catenin. 

## 7. Major highlights and Summary

Deimination of structural proteins of IF and IFAP:Influences their polymerization and interactionIncreases their solubility in low ionic strength bufferPromotes their proteolysis by several peptidasesPromotes their cross-linking by transglutaminases

Deimination and PADs target some arginine residues located at the accessible ends of the substrate, including head and tail regions of keratins, lamins and histones.

The activity of PADs is tightly controlled and the enzymes are “multilocked” in normal conditions.

Deimination and PADs are induced:By various exogenous environmental stimuli (chemical trauma, pressure and atmospheric dryness)By various chemical or biological agents (retinoic acid, calcium ionophores and inductors of apoptosis or NETosis)In many pathological contexts (rheumatoid arthritis, multiple sclerosis, Alzheimer’s disease, cancers, etc.)

Thus, it is now clear that deimination, beyond other post-translational modifications, must be taken into account—especially during biochemical, proteomic or peptidomic analyses. PADs and deimination play a definite role in the fate of cellular proteins, in particular for IF and their IFAP, and beyond on the various protein networks (cytoskeleton, fibrin network, etc.) and their dynamics. The active involvement of deimination and PADs in the extracellular compartment remains to be explored and could open up new perspectives in pathophysiology. In the near future, it will certainly be essential to decipher their effects on the actin cytoskeleton, microtubules and adhesion structures. Deimination, as with many post-translational modifications, can lead to a loss (or a gain) of affinity of some antibodies (in particular, monoclonal antibodies). Let us not lose sight of this in our analysis and interpretation. Finally, a schematic summary is presented in Figure 4 to illustrate each PAD in physiological or pathological contexts.

## Figures and Tables

**Figure 1 ijms-21-08746-f001:**
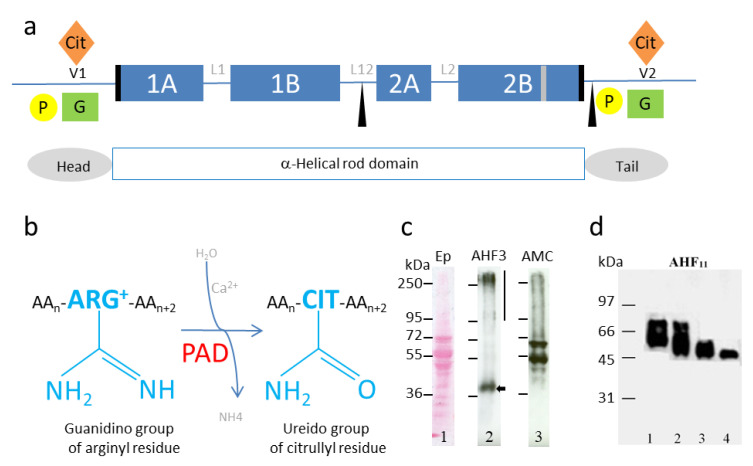
(**a**) Schematic representation of keratin structure, keratin major post-translational modifications and deamination reaction. P, phosphorylation, yellow dots; G, O-GlcNAcylation, green squares; Cit, citrullination or deimination, orange diamonds. Blue rectangles, 1A, 1B, 2A, 2B helicoidal domains constituting the α-helix rod domain; black boxes, helix initiation and termination motifs (Type I keratins do not contain any helix termination motifs); black triangles, caspase cleavage sites in Type I keratins; grey box, stutter region, variable amino-acid region inducing an imperfect coiled-coil arrangement [37]. L1, L12, L2, non-helicoidal linkers; Head and Tail, non-helicoidal terminal domains; V1/V2, domains highly variable in size. (**b**) Schematic representation of the irreversible reaction of deimination (or citrullination). AA, amino acid; Arg, arginine; Cit, citrulline (unconventional amino acid not encoded by a tRNA). (**c**) Deiminated epidermal keratins and filaggrin. Lane 1, Ponceau staining of a total protein extract of a human reconstructed epidermis. Note the keratin migration between 50 and 70 kDa, as expected. Lane 2, immunodetection of human (pro)filaggrin with the AHF3 specific monoclonal antibody [38], (filaggrin monomer, black arrow; filaggrin precursor, profilaggrin, black vertical line). Lane 3, deiminated proteins immunodetected with the anti-modified citrulline specific AMC antibody [17]. (**d**) Human recombinant His-Filaggrin deiminated by recombinant human PAD1, -2 or -3 (Lanes 1–3, respectively) or non-deiminated (control, Lane 4), as previously described [39]. A huge differential shift can be observed in the migration between the non-modified (Lane 4) and the deiminated (Lane 1–3) filaggrin. Filaggrin was immunodetected with AHF11 monoclonal antibody as previously described [39].

**Figure 2 ijms-21-08746-f002:**
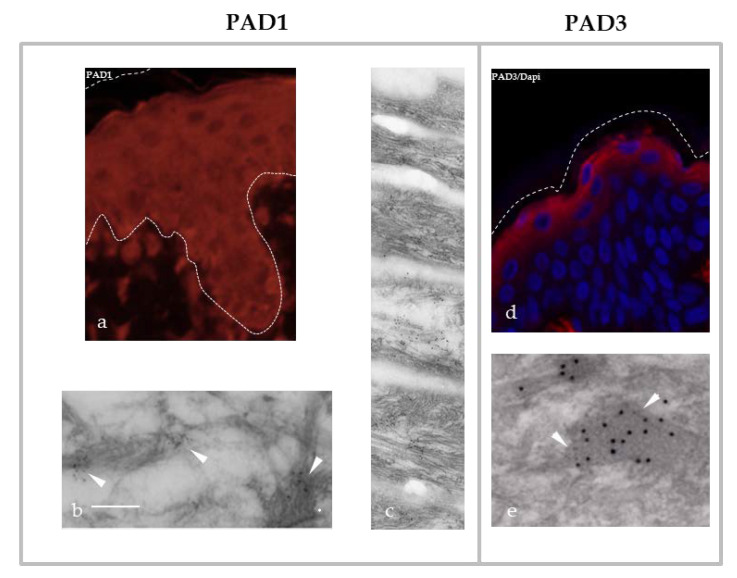
Localization of PAD1 (**a**–**c**) and PAD3 (**d**–**e**) in human epidermis. (**a**,**d**) Immunofluorescence and (**b**,**c**,**e**) immuno-electronic microscopy detection on normal human skin. Bar, 200 nm. The white dashed lines indicate either the dermo-epidermal junction (**a**) or the top of the *Stratum corneum* (**a**,**d**). White arrowheads point to the immunogold labeling of PAD1 on keratin IF (**b**) and PAD3 on keratohyalin granules (**e**).

**Figure 3 ijms-21-08746-f003:**
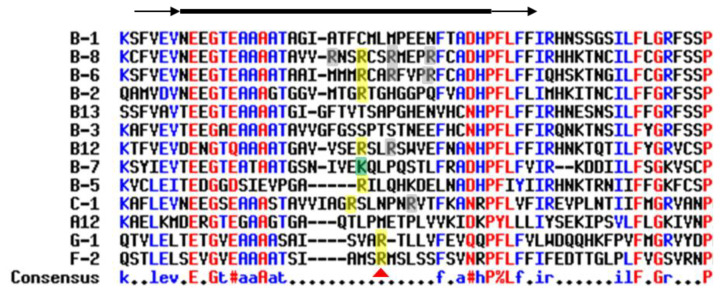
Multiple alignment of 13 serpin reactive center loops (RCL). Multiple alignment (Multalin) of the full primary sequence of 13 serpins was performed (available at http://multalin.toulouse.inra.fr/multalin/). At the top of the figure, RCL region is delimited by a thick black line and surrounded by beta-sheets (black arrows). Amino acids in red are conserved in all sequences and those in blue are partially conserved. Uniprot accession numbers of the 13 human serpins used are: A12 (Q8IW75), B1 (P30740), B2 (P05120), B3 (P29508), B5 (P36952), B6 (P35237), B7 (O75635), B8 (P50452), B12 (Q96P63), B13 (Q9UIV8), C1 (P01008), G1 (P05155) and F2 (P8697). The positively charged arginine (R) at position P1 of the RCL is indicated by a red triangle for serpin F2 and is highlighted in yellow for the others. Other arginines of the RCL are highlighted in grey, and lysine (K) in green.

**Figure 4 ijms-21-08746-f004:**
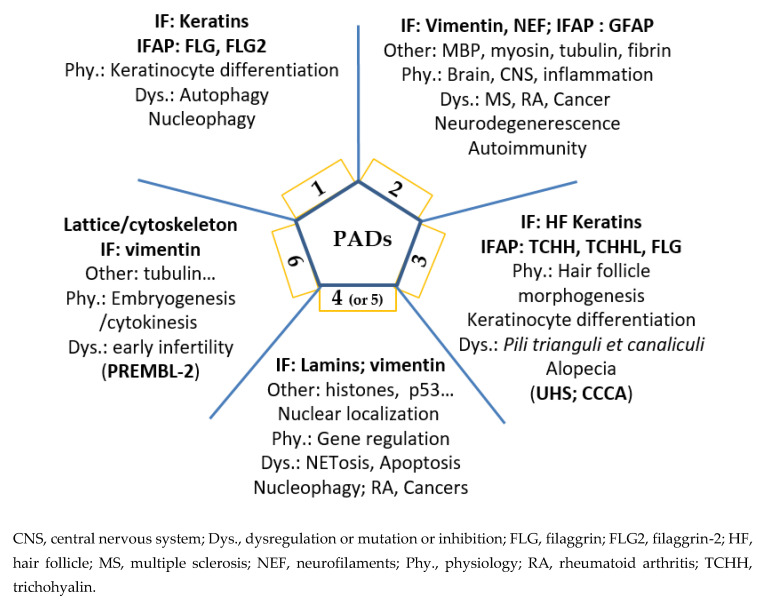
Schematic representation of the major involvements of PADs.

**Table 1 ijms-21-08746-t001:** Classification of intermediate filament proteins and some intermediate filament associated proteins (IFAP).

Proteins	Abbreviation	Uniprot Identifier	Type	Mass (kDa)	Cell Type/ Compartment	References ^1^
(Cyto)-keratin	CK or K	>50 genes	I (acidic)	40–60	Keratinocyte/cytosol *(epidermis, hair follicle, nail)*	[17,18,19]
			II (basic)	50–70		
Vimentin	VIM	P08670	III	54	Mesenchymal cell/cytosol	[20,21]
Desmin	DES	P17661	III	54	Muscular cell/sarcomere	/
Glial fibrillary acidic protein	GFAP	P14136	III	50	Glial cell, astrocyte/cytosol	[22]
Peripherin	PRPH	P14136	III	54	Peripheral neuron/cytosol	/
Neurofilaments	NF-L	P07196	IV	62	Neuronal axon/cytosol	/
	NF-M	P07197	IV	102	Neuronal axon/cytosol	
	NF-H	P12036	IV	112	Neuronal axon/cytosol	
Nestin	NES	P48681	IV	177	Neuroepithelial cell	/
Lamin A/C	LMNA	P02545	V	74	Nuclear lamina and nucleoplasm	[23]
Lamin B	LMNB1	P20700	V	66	Nuclear lamina and nucleoplasm	/
	LMNB2	Q03252	V	68	Nuclear lamina and nucleoplasm	
Filensin	BFSP1	Q12934	VI	75	Lens epithelial cell	/
Phakinin	BFSP2	Q13515	VI	46	Lens epithelial cell	/
(Pro)filaggrin	FLG	P20930	IFAP	>250/37	Granular keratinocyte	[17,18,19,22]
Trichohyalin	TCHH	Q07283	IFAP	>250	Hair follicle keratinocyte *(IRS, medulla)/*cytosol	[13,24,25,26]

^1^ References related to deimination of IF proteins and IFAP. IRS, inner root sheath.

**Table 2 ijms-21-08746-t002:** Examples of post-translational modifications (PTM) of intermediate filament proteins, intermediate filament associated proteins and PADs.

	Observed PTMs (√)															
Proteins	Phosphorylation	*O*-Linked Glycosylation	Ubiquitination	Sumoylation	Acetylation	Transamidation	Farnesylation	ADP-Ribosylation	Proteolysis	Cross-linked	*N*-Acylation	GTP-Ribosylation	Deamidation	Arg. Methylation	*S*-Palmitoylation	Deimination ^1^
Keratins	√	√	√	√	√	√										√
Lamins	√		√	√			√									√
Vimentin	√	√	√	√				√								√
GFAP	√															√
Neurofilaments	√	√	√													√
Desmin	√		√					√								√
(Pro)Filaggrin	√								√	√						√
TCHH	√								√	√						√
MBP ^2^	√							√	√		√	√	√	√		√
PADs	√														√ ^3^	√ ^4^

^1^ Deimination (or citrullination) is the transformation by PADs of an arginyl residue (positively charged) into a citrullyl residue (neutral) in a protein (or a peptide) substrate. This is not a free citrulline, therefore it is independent of the nitric oxide synthase pathway. ^2^ MBP is able to interact with microtubules and actin filaments (see also [27,28,29]). ^3^ Palmitoylation of Pad3 has been demonstrated. ^4^ Deimination of PAD1–4 refers to auto-deimination or auto-citrullination.

**Table 3 ijms-21-08746-t003:** *PADI* genes and human genetic disorders.

Genes	OMIM	Diseases or Human Gene Identification	Reference
*PADI1*	607934	*PADI1* cDNA cloning from a normal human keratinocyte library	[41]
*PADI*2	607935	*PADI2* cDNA cloning from a human cutaneous squamous cell carcinoma	[69]
*PADI*3	606755	*PADI3* cDNA cloning from cultured human keratinocytes	[70]
	191480	Uncombable hair syndrome (UHS)	[66]
	618352	Central centrifugal cicatricial alopecia (CCCA)	[67]
*PADI*4	605347	*PADI4* cDNA cloning from cultured human keratinocytes	[48]
	180300	Haplotypes associated with susceptibility to rheumatoid arthritis (RA)	[71,72,73]
*PADI*6	610363	*PADI6* cDNA cloning from a normal human ovary library	[40]
		Proteomic identification in mouse eggs (ePAD)	[74]
	617234	Preimplantation embryonic lethality-2 (PREMBL-2)	[75,76]

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
