# Peer review of "Deimination, Intermediate Filaments and Associated Proteins"

_ijms, 2020, doi:10.3390/ijms21228746_

Round 1
Reviewer 1 Report
Briot J et al. focused the role of PAD enzymes and citrullination in intermediate filaments and associated proteins. This review is substantial in content. The authors facilitated the understanding about this topic. Therefore, non-specialists in this field can be also comprehensible.
Author Response
Thank you for your positive comments.
Reviewer 2 Report
This review describes recent progress in the study of protein arginine deimination by peptidylarginine deiminases (PAD's). Especially, it summarizes the functions of PAD subtypes and their respective targets proteins in terms of physiology as well as pathophysiology. These are well written so that every readers can understand and obtain systematic information on recent advances in this research field. To me it looks it is worth being published in this journal.
Author Response
We would like to thank this Reviewer for his/her positive comments.
Reviewer 3 Report
Briot et al. show here in an invited review the relationship between “Deimination, Intermediate Filaments and Associated Proteins”.
They report very nicely on literature and contexts in thematic areas:
Deimination of structural proteins of intermediate filaments (IF) and associated proteins (IFAP), its influence on polymerization and interaction and the increased solubility in low ionic strength buffer. Furthermore, citrullination of IF and IFAP promotes their proteolysis by several peptidases and their cross-linking.
Citrullination and PADs target arginine residues at the ends of the substrates. While the activity of PADs is tightly controlled and the enzymes are "multilocked" in normal conditions, citrullination and PADs are induced by multiple exogenous environmental factors, including chemical and biological agents as also under disease conditions (e.g. MS, RA).
The review provides a very good overview, is clearly structured, written in good English, rich in content, and well documented and illustrated.
There is little to diminish enthusiasm. Only the Table2 and Figure3 would have deserved a better triggering and Figure3 should not look like it was cut out of a program with copy/paste but should be a bit more attractive. PAD1 and 2 in Figure 5 should also be in pentagons of the same size and in straight type without angles.
All in all a very successful work, which requires only minor improvements.
Author Response
Thank you for your positive comments and suggestions. All changes are listed below:
Line 65: Table 2 was modified as recommended.
Line 80: The legend of the new table 2 was completed by two new sentences: « ***, Palmitoylation of Pad3 has been demonstrated. » « ****, Deimination of PAD1-4 referes to auto-deimination or auto-citrullination. ».
Line 386: Figure 3 was modified, as requested.
Line 390: The legend of the new Figure 3 was completed by the added sentence: «At the top of the figure, RCL region is delimited by a thick black line and surrounded by beta-sheets (black arrows).».
Line 518: Figure 4 was modified, as requested.
Reviewer 4 Report
In this manuscript Julie Briot and colleagues review the topic “Deimination, Intermediate Filaments and Associated Proteins” with interesting insights about the role of this post-traslational modification in intermediate filaments (IFs) and associated proteins (IFAPs), in particular regarding their solubility, increased susceptibility to proteolysis, etc. Moreover, the coordinated action of deimination with the expression and enzymatic activity of specific Peptidylarginine deiminases (PADs) is remarkable, in particular if this PTM is somehow strongly related with the pathophysiology of autoimmune and degenerative diseases. This information is useful for newcomers in the field, and also for specialists in proteomics, which are not always aware of the potential effects of this PTM on the function of the identified proteins.
Author Response
Thank you for your positive reviewing.